# The Long Scientific Journey of Sirolimus (Rapamycin): From the Soil of Easter Island (Rapa Nui) to Applied Research and Clinical Trials on β-Thalassemia and Other Hemoglobinopathies

**DOI:** 10.3390/biology12091202

**Published:** 2023-09-02

**Authors:** Roberto Gambari, Cristina Zuccato, Lucia Carmela Cosenza, Matteo Zurlo, Jessica Gasparello, Alessia Finotti, Maria Rita Gamberini, Marco Prosdocimi

**Affiliations:** 1Center “Chiara Gemmo and Elio Zago” for the Research on Thalassemia, Department of Life Sciences and Biotechnology, University of Ferrara, 44121 Ferrara, Italy; cristina.zuccato@unife.it (C.Z.); gamberinimariarita@gmail.com (M.R.G.); 2Department of Life Sciences and Biotechnology, University of Ferrara, 44121 Ferrara, Italy; luciacarmela.cosenza@unife.it (L.C.C.); matteo.zurlo@unife.it (M.Z.); jessica.gasparello@unife.it (J.G.); 3Rare Partners S.r.L. Impresa Sociale, 20123 Milan, Italy; m.prosdocimi@rarepartners.org

**Keywords:** β-thalassemia, γ-globin genes, rapamycin (sirolimus), fetal hemoglobin, clinical trials

## Abstract

**Simple Summary:**

In this review article, we present the fascinating story of sirolimus (rapamycin), a drug known to be able to induce fetal hemoglobin, and for this reason of great interest for the treatment of β-thalassemia. In fact, high levels of fetal hemoglobin have been demonstrated to be beneficial for β-thalassemia patients. The story began in 1964, with METEI (Medical Expedition to Easter Island, Rapa Nui). During this expedition, samples of the soil from different parts of the island were collected and, from this material, an antibiotic-producing microorganism (*Streptomyces hygroscopicus*) was identified and rapamycin was extracted from the mycelium with organic solvents. The story continued with the finding that rapamycin was a very active anti-bacterial and anti-fungal agent. In addition, rapamycin was demonstrated to inhibit the cell growth of tumor cell lines. More importantly, rapamycin was found to be an immunosuppressive agent applicable to prevent kidney rejection after transplant. More recently, rapamycin was found to be a potent inducer of fetal hemoglobin both in vitro using cell lines, in vivo using experimental mice, and in patients treated with this compound. These studies were the basis for proposing clinical trials on β-thalassemia patients.

**Abstract:**

In this review article, we present the fascinating story of rapamycin (sirolimus), a drug able to induce γ-globin gene expression and increased production of fetal hemoglobin (HbF) in erythroid cells, including primary erythroid precursor cells (ErPCs) isolated from β-thalassemia patients. For this reason, rapamycin is considered of great interest for the treatment of β-thalassemia. In fact, high levels of HbF are known to be highly beneficial for β-thalassemia patients. The story of rapamycin discovery began in 1964, with METEI, the Medical Expedition to Easter Island (Rapa Nui). During this expedition, samples of the soil from different parts of the island were collected and, from this material, an antibiotic-producing microorganism (*Streptomyces hygroscopicus*) was identified. Rapamycin was extracted from the mycelium with organic solvents, isolated, and demonstrated to be very active as an anti-bacterial and anti-fungal agent. Later, rapamycin was demonstrated to inhibit the in vitro cell growth of tumor cell lines. More importantly, rapamycin was found to be an immunosuppressive agent applicable to prevent kidney rejection after transplantation. More recently, rapamycin was found to be a potent inducer of HbF both in vitro using ErPCs isolated from β-thalassemia patients, in vivo using experimental mice, and in patients treated with this compound. These studies were the basis for proposing clinical trials on β-thalassemia patients.

## 1. Introduction

Among hemoglobinopathies, the β-thalassemias are very frequent worldwide. These pathologies are caused by large genomic deletions within the β-like globin gene cluster or mutations within the gene encoding β-globin, causing the absence or decreased content of this protein in erythropoietic cells [1,2,3] with phenotypes ranging from asymptomatic (β-thalassemia trait or carrier) to clinically severe anemia, categorized as transfusion-dependent and non-transfusion dependent β-thalassemia (TDT and NTDT, respectively).

High production of fetal hemoglobin (HbF) has been demonstrated to be beneficial for β-thalassemia patients [4,5,6]. In fact, patients with rare forms of β-thalassemia, including those associated with hereditary persistence of fetal hemoglobin (HPFH), exhibit a relatively benign clinical course even if β-globin production is absent; this is due to the high quantity of γ-globin chain production, resulting in high HbF levels [7]. More recent pharmacogenomic studies have disclosed that the naturally higher production of HbF associated with well-characterized DNA polymorphisms (for instance, the XmnI polymorphism) improves the clinical course in a variety of β-thalassemia patients [8,9,10,11,12].

In this review article, we present the story of sirolimus (rapamycin), from its discovery to its validation as a fetal hemoglobin inducer and to its employment in applied research studies on hemoglobinopathies for which HbF induction is important (such as β-thalassemia and sickle-cell disease).

Interestingly, rapamycin gave the name to a family of proteins, collectively known as TORs (target of rapamycin), ubiquitously present and involved in many fundamental biological processes. As a consequence, it is not unexpected that rapamycin has multiple pharmacological effects. Accordingly, the involvement of rapamycin in other biological processes and pathologies will be also briefly presented and discussed.

## 2. From the METEI Expedition to the Discovery of Rapamycin (Sirolimus)

The fascinating story behind the discovery of rapamycin began during the Canadian Medical Expedition to Easter Island (METEI). A team of about 40 doctors and scientists participated in METEI [13,14,15]; they embarked in November 1964 on the Royal Canadian Navy’s H.M.C.S. Chief Scott in Halifax, Nova Scotia. They were bound for Easter Island (also known as Rapa Nui), a triangular-shaped speck in the South Pacific, located 2200 km from its nearest inhabited neighbor [13,14].

Key information on METEI can be found in the article by James A. Boutilier (Rapa Nui Journal, 1992) [13] and in a recently published book by Jacalyn Duffin (Stanley’s Dream: The Medical Expedition to Easter Island) [15].

METEI had several objectives and activities, both scientific, methodological, and infrastructure-oriented, all parts of an integrated medical study for the identification and evaluation of interconnected hereditary and environmental factors in the isolated native population of Easter Island. METEI activities were focused on investigations in the fields of applied ecology, sociology, anthropology, genetics, microbiology, and epidemiology, together with the development of methods of sampling procedures, and the collection and transport of blood and other biological material. A key objective of METEI was also to establish an Easter Island Biological Station, constituted of infrastructures and facilities for the assistance of the population in issues related to health and welfare.

Easter Island appeared to the organizers to be an excellent site for reaching the METEI objectives, since Rapa Nui, as it is known, is located in the southeast Pacific, 2300 miles (3700 km) off the coast of Chile, and is the most isolated inhabited island in the world, with an indigenous population (in December 1964) of 949 [13]. The only regular external contact of the islanders with the mainland was an annual supply vessel [13].

An additional objective of METEI was to identify and purify new antimicrobial agents. In particular, the microbiologists of the Easter Island expedition were deeply interested in understanding why the indigenous population of Easter Island, despite walking around barefoot, did not get tetanus [14].

During the expedition in 1964, the microbiologist Georges Nógrády collected soil samples from different parts of the island, in order to bring back material useful to identify anti-microbial agents responsible for the resistance of the inhabitants of Easter Island to tetanus [15]. Tetanus spores were not found in most of the collected samples; however, the soil samples were not discarded by Dr. Nógrády, and were transferred to scientists working at Ayerst Pharmaceuticals on medicinal compounds made by bacteria [16].

Within a soil sample obtained by the METEI expedition to Easter Island, *Streptomyces hygroscopicus*, an antibody-producing microorganism, was identified. From the mycelium, a lipophilic macrolide, named rapamycin, was extracted with organic solvent and isolated in crystalline form [17]. This molecule was further characterized with respect to its biological activity, and was found to be active against *Candida albicans*, *Microsporum gypseum*, and *Trichophyton granulosum*. Studies conducted within the National Cancer Institute (NCI) Developmental Therapeutics Program demonstrated rapamycin-mediated inhibition of cell growth in tumor cell lines [18,19]; this finding stimulated further efforts to explore the rapamycin mechanism responsible for these anti-tumor activities. These studies, together with others pointing out the relevant bioactivities of rapamycin, launched a billion-dollar drug industrial activity and made possible biomedical research in a major field of investigation [20].

## 3. Rapamycin: Biomedical Applications

The chemical structure of rapamycin is presented in Figure 1A. Rapamycin binds and inhibits the mechanistic Target Of Rapamycin (mTOR) [21], a serine/threonine-specific protein kinase regulating cell growth, proliferation, survival, mobility, and angiogenesis [22,23]. More in detail, rapamycin binds to the cytoplasmic receptor FK506-binding protein-12 (FKBP12), generating an immunosuppressive complex that is able to bind and inhibit mTOR [22,23]. These interactions play important roles in regulating downstream signaling pathways affecting cell survival, such as the phosphatidylinositol-3 kinase (PI3K)/Akt signaling pathway [24].

One biomedical application that was extensively validated in pre-clinical and clinical studies was the prevention of transplant rejection, due to the immunomodulatory action of mTOR inhibitors [25]. After intense research activity, it became clear that the mTOR signaling pathway is deeply involved in the cellular responses to environmental changes in nutrients and oxygen status, and is a key regulator of cell growth, translation, cell survival, and other important biological processes, such as autophagy and cytoskeletal rearrangements [26].

Since the dysregulation of the mTOR pathway is implicated in several diseases (in particular neoplastic disorders), mTOR inhibitors, such as rapamycin and rapamycin analogs (rapalogs), exhibit a number of useful actions. For instance, rapamycin (sirolimus) and rapalogs have been employed and/or studied for the treatment of patients undergoing organ allotransplantation [27,28,29,30,31], and for patients affected by systemic lupus erythematosus (SLE) [32], lymphangioleiomyomatosis (LAM) [33], tuberous sclerosis complex [34], recurrent meningioma [35], pancreatic neuroendocrine tumors (NET) [36], advanced differentiated thyroid cancers [37], advanced breast cancer [38], B-cell lymphomas [39], and metastatic renal cell carcinoma [40].

## 4. Rapamycin in Organ and Tissue Transplantation

The most important and studied clinical applications of rapamycin and rapalogs are based on their immunosuppressant activity in the treatment of patients undergoing organ allotransplantation. Several experimental studies and review articles are available on this issue [25,41,42,43,44,45,46]. Sirolimus (rapamycin) is approved by Food and Drug Administration and other regulatory authorities to prevent organ rejection in patients receiving renal transplantation [44,45,46]. Moreover, a high number of clinical studies have been published using sirolimus as bioactive molecules in the treatment of patients who underwent transplantation of the lung [47], heart [48], pancreas [49], liver [50], intestine [51], cornea [52], and bone marrow [53]. Consequently, sirolimus (rapamycin) is of great interest for several pathologies in which transplantation is a clinical option, including (but not restricted to) solid cancers (for instance liver transplantation in hepatocellular carcinoma, or kidney transplantation in renal cancers) [54,55], leukemia, lymphoma and other blood cancers (for instance bone marrow transplantation in leukemia) [56], cystic fibrosis (lung transplantation) [57], serine/threonine kinase 4 (STK4) deficiency, characterized by recurrent bacterial, viral, and fungal infections (allogeneic hematopoietic stem cell transplantation, HSCT) [58], and hematological diseases (bone marrow transplantation) [59]. In clinical allotransplantation, the long-term efficacy of rapamycin and other mTOR inhibitors has been firmly established, despite the fact that adverse events have been reported, such as the inhibition of wound healing, buccal ulceration, anemia, hyperglycemia, dyslipidemia, and thrombocytopenia [43,60]. However, the general agreement is that the benefits of the use of mTOR inhibitors in allotransplantation exceed the adverse effects [43]. In order to optimize the immunosuppressant therapy in kidney-transplanted patients, pharmacogenetics studies have been considered. In this respect, Urzì Brancati et al. proposed that the characterization of the polymorphisms in CYP3A5, CYP3A4, ABCB1, and UGT1A9 genes could be a strategy to select the ideal dosage for each patient [61].

Finally, we should consider that the positive impact of allotransplantation in treating human pathologies is deeply mitigated by the difficulty in finding organ donors compatible with the recipients. In order to facilitate this time-consuming activity, xenotransplantation might be considered as a temporary remedy [62]. Recently published reports demonstrated the possible use of sirolimus and mTOR inhibitors for xenotransplantation [63].

## 5. Rapamycin for Longevity?

The first sets of data supporting the hypothesis that rapamycin might affect longevity were produced in studies on invertebrates. For instance, in 2003, Vellai et al. reported that TOR mutations were associated with an increase in the lifespan of *Caenorhabditis elegans* [64]. In agreement with this study, other research groups demonstrated that mutations in TOR increased the lifespan of yeast [65] and Drosophila [66]. In agreement with the evidence that the inhibition of TOR signaling was associated with an increase in lifespan, the NIA Intervention Testing Program proposed to test the effect of feeding rapamycin to mice [67]. The first data of this study were reported by Harrison et al. [67] and demonstrated that feeding rapamycin was effective in increasing the lifespan of mice. The increase in lifespan was observed also when rapamycin was given to mice later in life [68].

Rapamycin was employed by different research groups using different dosages in different mouse strains with similar effects in increasing lifespan [68]. Starting from these pre-clinical studies concurrently demonstrating that mTOR is a key modulator of aging (but also of age-related diseases), Kaeberlein et al. evaluated the potential of rapamycin use to promote health span in human adults [69]. They collected data from 333 adults with a history of off-label use of rapamycin and made a comparison with data collected from 172 adults who had never used rapamycin. The results obtained further supported interventional studies based on rapamycin to improve the quality of life, especially in the elderly [69].

## 6. Rapamycin: Effects on the Immune System

An unexpected finding reported by different research groups was that mTOR inhibitors improve vaccine responses, especially in the elderly. This activity was surprising, and saw these molecules defined as “immune suppressant”. An example of this property of mTOR inhibitors is the boosting of the vaccine against the influenza virus [70,71]. Accordingly, mTOR inhibitors might be considered for improving the overall health of aged populations [72]. In addition to this line of research results, TOR inhibitors have been proposed to enhance CD8+ effector memory T cell function [73,74]. Collectively, there is agreement on the fact that boosting vaccination of elderly and fragile people as well as memory T cell function, in general, can be obtained with the use of mTOR inhibitors (including rapamycin) [75,76,77].

## 7. Testing Rapamycin (Sirolimus) on Erythroid Cells: Induction of Fetal Hemoglobin

An important step for the possible application of rapamycin in hematological disease was the demonstration that this molecule is able to induce erythroid differentiation of K562 cells. This cell line was obtained by Lozzio and Lozzio from the pleural effusion of a patient with chronic myeloid leukemia (CML) in blast crisis [78]. The reasons for using this cell line were based on the studies reporting the very interesting observation that the K562 cell line shares properties in common with erythroid cells [79,80,81]. In particular, K562 cells contain glycophorin A [80] and spectrin [81] in their membranes and synthesize minute amounts of hemoglobin. Key studies that demonstrated the usefulness of K562 cells for the screening of HbF inducers were published by Rutherford et al. [82], who investigated the capacity of these cells to differentiate following treatment with hemin, producing large amounts of hemoglobin, mainly the embryo-fetal hemoglobins Hb Gower 1 (α_2_ε_2_), Hb Portland (ζ_2_γ_2_), and to a lesser extent HbF (α_2_γ_2_). Accordingly, the erythroleukemia K562 cell line was used in several studies for the first screening of potential inducers of the expression of γ-globin genes and accumulation of fetal hemoglobin [83,84,85,86,87,88].

With respect to the clinical impact on β-thalassemia and sickle-cell disease (SCD) patients, sirolimus has been considered a potentially useful drug since its ability to induce fetal hemoglobin has been demonstrated in erythroid cells isolated from β-thalassemia and SCD patients [89,90,91,92] (see Figure 2).

The relevance of these studies is due to the fact that novel HbF inducers are still greatly needed. In fact, hydroxyurea (HU) is frequently used with positive results on β-thalassemia patients, but has potential adverse effects and exhibits efficacy in only a subset of patients [103,104,105,106]. Furthermore, patients might become HU-resistant after long-term treatment.

As for the mechanism of action of sirolimus and rapalogs for HbF induction, Bianchi et al. [107] and Finotti et al. [108] have proposed mTOR as a key regulator of erythroid differentiation and globin gene expression. In this context, it was found that hypophosphorylation of α-p-S6 ribosomal protein and 4E-BP-1 is associated with sirolimus-mediated erythroid induction. Furthermore, the inactivation of both 4E-BP1 and p70-S6K are sufficient steps for the induction of erythroid differentiation [107]. Interestingly, other HbF inducers (i.e., mithramycin, which targets the mTOR regulator raptor) specifically inhibit the mTOR pathway. Finally, HbF induction can be potentiated by siRNAs and/or microRNAs targeting mTOR, or other members of the mTOR pathways [107,108].

## 8. Testing Rapamycin on Animal Models: Supporting Evidences for a Possible Role in the Therapy of β-Thalassemia

The effects of sirolimus in inducing HbF in vivo have been sustained by several studies using animal model systems. In 2014, Zhang et al. published a paper providing data on the redox and metabolic regulation of erythropoiesis [93]. Among the several experiments that were designed and carried out, Zhang et al. studied the effects of rapamycin treatment in Th3/+ mice, considered a good model to study ineffective erythropoiesis. Increases in red blood cell counts and hemoglobin levels were observed in mice following treatment with rapamycin. These results supported the hypothesis that the administration of rapamycin would be of help in patients with ineffective erythropoiesis.

In a murine model system of sickle-cell disease, Khaibullina et al. verified that rapamycin treatment significantly increased γ-globin mRNA and HbF levels [94]. In another report, Wang et al. confirmed that mTOR inhibition would improve anemia [95]. Erythrocyte count, hemoglobin, and hematocrit were all significantly increased in mice treated with the mTOR inhibitor INK128 [95]. Prolonged erythrocyte lifespan, reduced spleen size, and reduced iron accumulation in the kidney and liver were observed in sirolimus-treated treated SCD mice [95].

More recently, Lechauve et al. [96] proposed an alternative mechanism explaining the sirolimus-mediated improvement of the phenotype of thalassemic mice: inhibition of mTOR can activate ULK1 and autophagy, thereby causing a significant reduction in the excess of free α-globin accumulation. This was associated with a reduction in ineffective erythropoiesis and a longer lifespan of red blood cells [96]. All these effects are very important for the treatment of patients with β-thalassemia.

## 9. IP Protection and Orphan Drug Designation

When our group originally explored the possibility of using rapamycin to treat β-thalassemia [89,90,91], knowledge of the action of the drug on erythroid cells was still largely incomplete. On the other hand, many experiments were already designed to characterize agents capable of augmenting HbF levels in humans [109]. Along this line, our group found that many agents, some of them derived from plants, were indeed able to exert such action in vitro. Rapamycin was soon validated as a very interesting product, allowing several patent applications followed by grants. Quoting the most relevant documents, patent protection has been granted by EP1521578B1 and by US 7541380-B2 (see Figure 2). The granting of the patents was considered a fundamental step of a joint collaboration formally started in 2011 and involving two patient associations (AVLT and ALT), the University of Ferrara, and the small non-profit company Rare-Partners (RP). During this collaborative effort, it was recognized that, for a repurposed drug (in this case rapamycin/sirolimus for β-thalassemia and sickle cell disease), an important step was to obtain Orphan Drug Designation (ODD) from regulatory authorities. On the basis of the data published by Gambari and coworkers, as well as by other scientists [103], Rare Partners applied for ODD for β-thalassemia and for sickle cell disease in both Europe and the United States. ODDs were granted in 2015 and 2016 for beta-thalassemia and in 2017 and 2018 for sickle cell disease. Finally, the EMA scientific opinion was also requested in order to design the clinical trials in β-thalassemia (NCT03877809 and NCT04247750, one of which (NCT03877809) has been recently concluded and the other is still ongoing. [97,98].

## 10. mTOR Inhibitors and Clinical Case Reports: First Evidences on In Vivo Effects on HbF

The in vivo effects of mTOR inhibitors on patients affected by hemoglobinopathies were first studied analyzing HbF production in rapalogs-treated patients who underwent kidney transplantation. Gaudre et al. [99] reported the case of an SCD patient who received a kidney transplant and was treated with the mTOR inhibitor everolimus. After 10 months of everolimus treatment, the patient’s HbF levels were found to be dramatically increased, supporting the concept that rapalogs might be able to induce HbF production in vivo [99]. Similar results were reported by Al-Khatti et al. using sirolimus on SCD patients [100]. In addition, a further increase in HbF levels was found when sirolimus and hydroxyurea were combined [100].

Despite the fact that these were a few case reports, the results obtained were important, encouraging the organization of the first pilot clinical trials using sirolimus on β-thalassemia patients.

## 11. The Sirthalaclin and Thala-Rap Clinical Trials on β-Thalassemia

Two clinical trials have been recently conducted on ß-thalassemia patients (Sirthalaclin, NCT03877809, and Thala-Rap, NCT04247750) using low dosages of rapamycin [97].

The main objective of these two phase II studies was to verify the efficacy of rapamycin (sirolimus) as an in vivo HbF inducer. The first protocol (NCT 03877809) was a single-center clinical trial conducted at the Thalassemia Centre of Azienda Ospedaliera-Universitaria S. Anna (Ferrara, Italy) on patients with β^+^/β^+^ and β^+^/β^0^ genotypes [97]. The second protocol (NCT 04247750) was a multicenter study conducted in Ferrara, Firenze, and Pisa on β-thalassemia patients with β^0^/β^0^ and β^+^/β^0^ genotypes [97].

The main results of the first trial have been recently published [98] and the data of the second trial are currently being analyzed.

The results obtained during the sirolimus-based NCT03877809 clinical trial [98] demonstrate that γ-globin mRNA increased in the blood and in the erythroid precursor cells ErPCs) isolated from treated β-thalassemia patients. In this trial, sirolimus was found to influence erythropoiesis and to reduce biochemical markers associated with ineffective erythropoiesis (for instance, the excess of free α-globin chains, and the content of bilirubin, soluble transferrin receptor, and ferritin) [98].

## 12. Rapamycin and Biomedical Applications in Hematology: The Journey Is Not over Yet

Considering the promising results of the NCT03877809 clinical trial [98], the rapamycin journey is not over yet; in fact, further larger clinical trials are warranted, possibly including testing of the drug in patients with less severe forms of the disease and exploring combination therapies (see Figure 2).

In this context, Zuccato et al. [101] found that sirolimus synergizes with the Cinchona alkaloids cinchonidine and quinidine for the maximal induction of HbF in erythroid progenitor cells isolated from β-thalassemia patients [101], suggesting that combined treatment should be considered in future experiments. Moreover, the potential synergistic activity of sirolimus should be verified when the combination protocols are designed with inhibitors of transcription factors down-regulating γ-globin gene expression. In this context, Gasparello et al. demonstrated a possible use of molecules mimicking the miR-210-3p activity for suppression of the γ-globin gene repressor BCL11A [110].

The combined treatments should also include novel genomic approaches, including gene therapy and genomic editing (GE). For instance, Cosenza et al. were recently able to demonstrate that the de novo production of adult hemoglobin (HbA) using CRISPR-Cas9 gene editing [111] can be combined with rapamycin-mediated HbF induction [102] (Figure 3).

The data obtained by Cosenza et al. [106] demonstrated that the maximum level of production of HbA and HbF was obtained in GE-corrected, rapamycin-induced erythroid progenitors isolated from the studied β^0^39-thalassemia patients [102].

Moreover, the sirolimus journey should be considered to verify its effects on patients with sickle-cell disease (SCD). While red blood cells are usually round and flexible, in SCD, some of them are shaped like sickles or crescent moons [112]. This is caused by the production of sickle hemoglobin, with a high tendency to precipitate, causing sickle cells to become rigid and sticky, which can slow or even block blood flow [112]. The finding of the anti-sickling property of HbF is the rationale for proposing sirolimus as a possible drug for SCD, considering its validated ability to induce HbF. Therefore, sirolimus is expected to be employed in clinical trials for SCD patients.

A final comment is on DNA polymorphisms associated with HbF induction. This is a relevant issue that is expected to impact the criteria for recruiting patients for sirolimus-based treatments. In this context, when the relationship between the rapamycin-induction of HbF and DNA polymorphisms was analyzed, the rs368698783 LYAR (G>A) [113,114] and the XmnI polymorphisms [115] displayed a high association. A low relationship was found when MYB rs9399137, BCL11A rs14227407, and BCL11A rs10189857 were analyzed [116]. A more extensive pharmacogenomic analysis including other HbF-associated polymorphisms will be necessary to determine the polymorphism displaying the highest association with rapamycin-mediated HbF induction. This will be a key achievement for personalized treatments in precision medicine of β-thalassemia and sickle-cell disease.

## 13. Conclusions

In this review, we presented the fascinating story of rapamycin (sirolimus), from its discovery, to its application in biomedicine, and to its use in recent clinical trials. The interest in rapamycin for β-thalassemia is based on the fact that this drug is able to induce γ-globin gene expression and increased production of fetal hemoglobin (HbF) in erythroid cells, including primary erythroid precursor cells (ErPCs) isolated from β-thalassemia patients. For this reason, the conclusion of several studies supports the concept that rapamycin should be considered of great interest in the treatment of β-thalassemia. In fact, high levels of HbF are known to be highly beneficial for β-thalassemia patients. Several studies have concluded that rapamycin is a potent inducer of HbF both in vitro using ErPCs isolated from β-thalassemia patients, in vivo using experimental mice, and in patients treated with this compound. These studies were the basis for proposing clinical trials on β-thalassemia patients. In future studies, we expect sirolimus to be tested in combination with other HbF inducers, with gene editing approaches and in clinical trials for sickle-cell disease (SCD).

In conclusion, even if many years elapsed from the beginning of the story and the isolation of rapamycin, many pages still deserve to be written. Many scientists, including ourselves, have written sentences and performed studies (Figure 2); let us see if and when a conclusive page will be produced.

## Figures and Tables

**Figure 1 biology-12-01202-f001:**
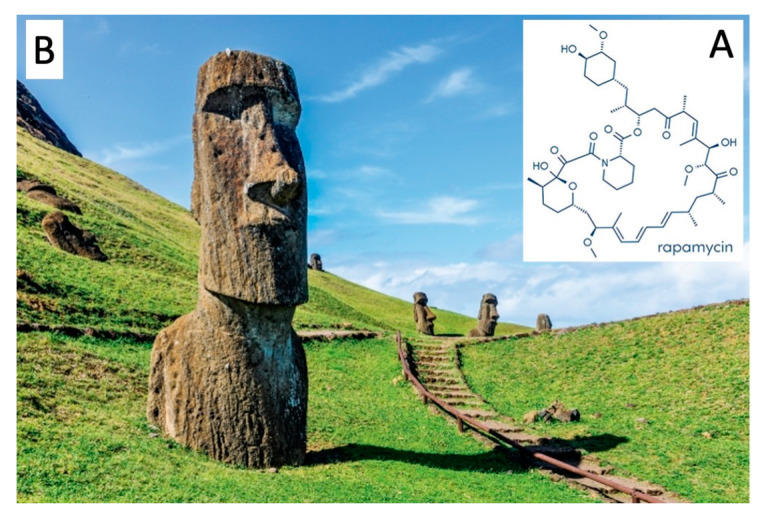
Structure of sirolimus (rapamycin (**A**)) isolated from soil samples of Easter Island (**B**) (credit: Shutterstock) [13] (https://www.shutterstock.com/it/search/rapa-nui: images unprotected by copyright, accessed on 31 December 2022).

**Figure 2 biology-12-01202-f002:**
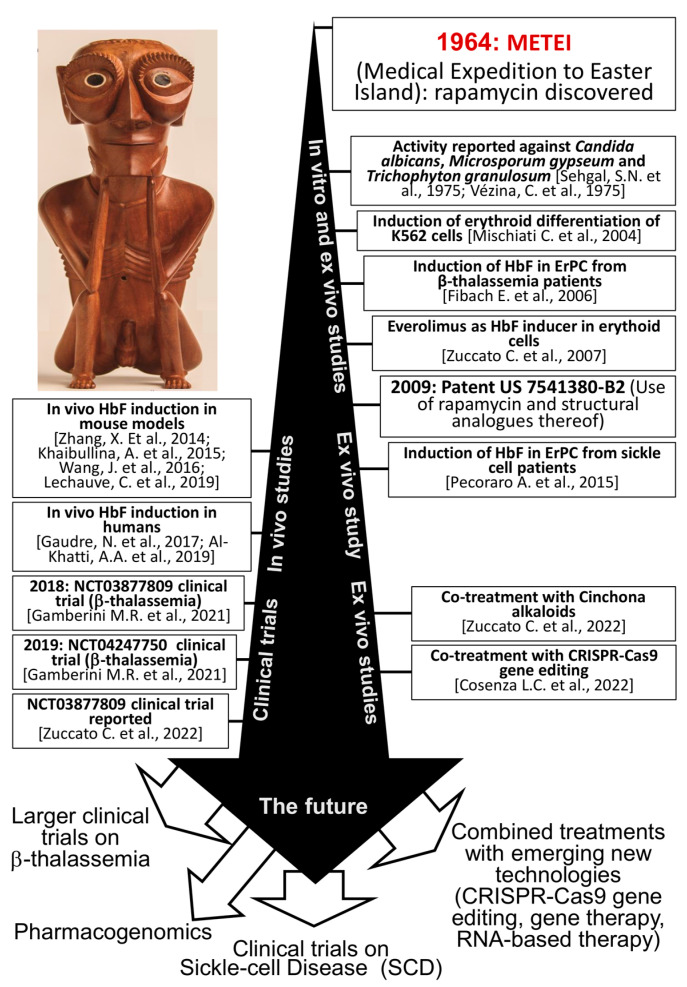
Phases of the journey driving sirolimus from discovery to clinical trials in hemoglobin disorders; the image on the top-left side of the panel was adapted with permission by the “Rapa Nui Museum” and was created by Tomás Tuki Tepano (https://www.museorapanui.gob.cl/noticias/mahana-o-te-hauhaa-henua; accessed on 28 May 2017); permission obtained on 13 January 2023. The indicated research activities have been taken from the following studies: Sehgal, et al., 1975 [16], Vézina et al., 1975 [17], Mischiati et al., 2004 [89], Zuccato et al., 2007 [90], Fibach et al., 2006 [91], Pecoraro et al., 2015 [92], Zhang, X. et al., 2014 [93]; Khaibullina, A. et al., 2015 [94], Wang, J. et al., 2016 [95], Lechauve, C. et al., 2019 [96], Gamberini et al., 2021 [97], Zuccato et al., 2022 [98], Gaudre, N. et al., 2017 [99]; Al-Khatti, A.A. et al., 2019 [100], Zuccato C. et al., 2022 [101], Cosenza L.C. et al., 2022 [102].

**Figure 3 biology-12-01202-f003:**
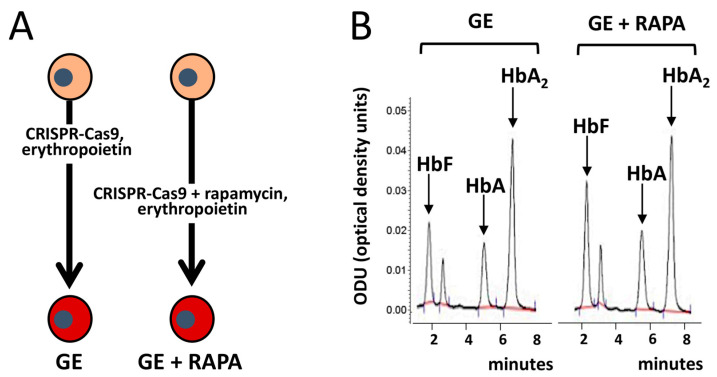
Co-treatment of ErPCs from β-thalassemia patients with rapamycin (RAPA) and a CRISPR-Cas9-based gene editing (GE) protocol for correction of the β^0^39-globin gene mutation. (**A**). Pictorial representation of the combined experimental protocols. (**B**). Demonstration that in (GE + RAPA)-treated cells HbF is increased, together with the “de novo” production of HbA. Modified from Cosenza et al., with permission (copyright can be found at https://www.mdpi.com/2073-4425/13/10/1727, accessed on 3 July 2023) [102].

## Data Availability

Not applicable.

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
