# Peer review of "The Long Scientific Journey of Sirolimus (Rapamycin): From the Soil of Easter Island (Rapa Nui) to Applied Research and Clinical Trials on β-Thalassemia and Other Hemoglobinopathies"

_biology, 2023, doi:10.3390/biology12091202_

Round 1

Reviewer 1 Report

I reviewed with great interest the article by Ferrara colleagues and I sincerely thanks for the opportunity to learn the Rapa story. The paper is interesting and very well written. Paper tells a story not unusual in modern medicine and clinical research: novel drugs = novel clinical opportunities for pur patients.

To improve the paper, I suggest the following:

Paper is not only on thalassemia (this journal is a general biology journal) but discloses the story of rapamycin in various diseases. I would omit thalassemia from the title, shorten paragraphs related to thalassemia and include a specific paragraph on organ-tissue transplantation that is one of the most common use of rapamycin (even in hemopoietic cell transplantation). Particularly I would shorten paragraph 7, 8 (case reports are not useful) and 9.

Paragraph 10 can be extended including main results of the published clinical trial.

Paragraph 11 is definitely too long.

Page 6: OHU definitely does’ work in thalassemia.

I do not like figure 2, can be omitted.

Too many references

Author Response

REPLY to Reviewer #1

General comments. I reviewed with great interest the article by Ferrara colleagues and I sincerely thanks for the opportunity to learn the Rapa story. The paper is interesting and very well written. Paper tells a story not unusual in modern medicine and clinical research: novel drugs = novel clinical opportunities for our patients. To improve the paper, I suggest the following.

Answer. We thank the reviewer for her(his) positive comments and we hope to have followed her(his) suggestions properly.

Point 1. Paper is not only on thalassemia (this journal is a general biology journal) but discloses the story of rapamycin in various diseases. I would omit thalassemia from the title, shorten paragraphs related to thalassemia and include a specific paragraph on organ-tissue transplantation that is one of the most common use of rapamycin (even in hemopoietic cell transplantation). Particularly I would shorten paragraph 7, 8 (case reports are not useful) and 9.

Answer. We fully agree that BIOLOGY Journal is interested in research studies covering a wide range of pathologies. However, we have been invited by the Journal to participate to the Special Issue "Thalassemia Research: Focus on Novel Molecular Insights and Clinical Perspectives" (Guest Editors: Michela Grosso and Dr. Alexandros Makis. For this reason, we would like to maintain the focus on thalassemia and hemoglobinopathies for which HbF induction is beneficial. In order to meet the reviewer’s suggestions we slightly changed the Title to: The long scientific journey of sirolimus (rapamycin): from the soil of Easter Island (Rapa Nui) to applied research and clinical trials on β-Thalassemia and other hemoglobinopathies” and explained the aim of the review by implementing the sentence “In this review article, we present the story of sirolimus (rapamycin), from its discovery to its validation as fetal hemoglobin inducer and to its employment in applied research studies on hemoglobinopathies for which HbF induction is important (such as β-thalassemia and sickle-cell disease)” (page 2, lines 60-62). We also included the short sentence “Accordingly, the involvement of rapamycin in other biological processes and pathologies will be also briefly presented and discussed” (page 2, lines 66-67) in order to follow the point raised about the multiple applications of rapamycin treatments. Finally, we fully agree that the review will be much improved by the inclusion of a section on organ-tissue transplantation (that is one of the most common uses of rapamycin). To this aim section #4 (Rapamycin in organ and tissue transplantation was included (page 4 and 5, lines 150-179) and references 41-63 were added. Interestingly, rapamycin has been also proposed for xenotransplantation (ref. 62 and 63). All the other references have been accordingly re-numbered.

We have reconsidered paragraphs 7, 8, and 9. As far as the observation that “case reports are not useful”, we clarified this point by adding the sentence “Despite the fact that these were a few case reports, the results obtained were important, encouraging the organization of the first pilot clinical trials using sirolimus on β-thalassemia patients” (page 7, lines 297-299).

Point 2. Paragraph 10 can be extended including main results of the published clinical trial.

Answer. We have extended this paragraph (now chapter 11) by including the sentence “The results obtained during the sirolimus-based NCT03877809 clinical trial [90] demonstrate that γ-globin mRNA increased …. the excess of free α-globin chains, and the content of bilirubin, soluble transferrin receptor, and ferritin) [105]” (page 8, lines 311-316). We would like to inform the reviewer that the second trial (THALA-RAP) confirms the key results of the Sirthalaclin NCT03877809 trial.

Point 3. Paragraph 11 is definitely too long.

Answer. We have shortened Paragraph 11 (now paragraph 12). In particular, we have shortened the part focusing on combined treatments with gene editing and future clinical trials on SCD.

Point 4. Page 6: OHU definitely does’ work in thalassemia.

Answer. We agree, but we still believe that additional HbF inducers are needed. We modified the sentence as follows: “The relevance of these studies is due to the fact that novel HbF inducers are still highly needed. In fact, hydroxyurea (HU) is frequently used with positive results on b-thalassemia patients but has potential adverse effects and exhibits efficacy in only a subset of patients [78-81]. Furthermore, patients might become HU resistant after long-term treatment.” (page 6, lines 229-233).

Point 5. I do not like figure 2, can be omitted.

Answer. We have reorganized Figure 2, which was overcrowded of information. We maintained the key scientific achievements concerning the application of rapamycin in hematological diseases. We removed all the information regarding technological transfer (that is however present in Paragraph #11) with the exception of the key US patent 7541380. We think that Figure 2 might be helpful to the reader and we hope that the new version will be considered acceptable.

Point 6. Too many references.

Answer. The following references have been removed (numbering of the original manuscript):

  1. Macian F. Autophagy in T Cell Function and Aging. Front Cell Dev Biol 2019;7:213.
  2. Gahmberg CG, Andersson LC. K562--a human leukemia cell line with erythroid features. Semin Hematol. 1981;18(1):72-7.
  3. Rutherford T, Clegg JB, Higgs DR, Jones RW, Thompson J, Weatherall DJ. Embryonic erythroid differentiation in the human leukemic cell line K562. Proc Natl Acad Sci U S A. 1981;78(1):348-52.
  4. Wang D, Si S, Wang Q, Luo G, Du Q, Liang Q, Guo X, Zhang G, Feng J, Leng Z. MiR-27a Promotes Hemin-Induced Erythroid Differentiation of K562 Cells by Targeting CDC25B. Cell Physiol Biochem. 2018;46(1):365-374.
  5. Voskou S, Phylactides M, Afantitis A, Melagraki G, Tsoumanis A, Koutentis PA, Mitsidi T, Mirallai SI, Kleanthous M. MS-275 Chemical Analogues Promote Hemoglobin Production and Erythroid Differentiation of K562 Cells. Hemoglobin. 2019 Mar;43(2):116-121.
  6. Khan F, Ali H, Musharraf SG. Tenofovir disoproxil fumarate induces fetal hemoglobin production in K562 cells and β-YAC transgenic mice: A therapeutic approach for γ-globin induction. Exp Cell Res. 2020;394(2):112168.
  7. Iftikhar F, Khan MBN, Musharraf SG. Monoterpenes as therapeutic candidates to induce fetal hemoglobin synthesis and up-regulation of gamma-globin gene: An in vitro and in vivo investigation. Eur J Pharmacol. 2021;891:173700.
  8. Ali H, Khan F, Musharraf SG. Cilostazol-mediated reversion of γ-globin silencing is associated with a high level of HbF production: A potential therapeutic candidate for β-globin disorders. Biomed Pharmacother. 2021;142:112058.
  9. Alipour M, Nasiri N, Kazemi F, Zare F, Sharifzadeh S. Resveratrol plus low-dose hydroxyurea compared to high-dose hydroxyurea alone is more effective in γ-globin gene expression and ROS reduction in K562 cells. Nat Prod Res. 2022; 1:1-5.
  10. Sebastiani P, Steinberg MH. Fetal hemoglobin per erythrocyte (HbF/F-cell) after gene therapy for sickle cell anemia. Am J Hematol. 2023 Feb;98(2):E32-E34. doi: 10.1002/ajh.26791.
  11. Menzel S.; Jiang J., Silver N.; Gallagher J.; Cunningham J.; Surdulescu G.; Lathrop M.; Farrall M.; Spector T.D.; Thein S.L. The HBS1L-MYB intergenic region on chromosome 6q23.3 influences erythrocyte, platelet, and monocyte counts in hu-mans. Blood J. Am. Soc. Hematol. 2007;105:3624–3626.
  12. Prasing W.; Mekki C.; Traisathit P.; Pissard S.; Pornprasert S. Genotyping of BCL11A and HBS1L-MYB Single Nucleo-tide Polymorphisms in β-thalassemia/HbE and Homozygous HbE Subjects with Low and High Levels of HbF. Walailak J. Sci. Technol. 2018;15:627–636.

The total number of references is still high, due to the fact that we have included chapter #4 (rapamycin and organ/tissue transplantation).

We thank the reviewer and we hope that the changes included following the suggestions will be considered acceptable.

Reviewer 2 Report

This is an interesting review on the history of rapamycin and its potential use as a de-repressor of gamma globin/fetal hemoglobin.

1. There are number of minor proofreading/grammar/usage issues, several of which are pointed out below.

2. Many authors prefer to use "mechanistic Target Of Rapamycin" as the elaboration of "mTOR" rather than "mammalian", because it is similarly applicable to non-mammalian or mammalian cell studies.

3. On line 53, the clinical syndrome to which the authors refer is more correctly designated "hereditary persistence of fetal hemoglobin", which more accurately correspondence with the acronym HPFH.

The manuscript would benefit from improved proofreading. There are at least several minor errors of English usage. Some are pointed out below:

Line 97: Should say “were transferred”

Line 105: Should say “rapamycin-mediated”

Line 108: Should say “others”

Line 243: Should say “pointed” instead of “spotted”

Line 278: Should say “being analyzed” instead of “under elaboration”

Line 283: Should say “influence”

Line 321: Should say “proposing”

Author Response

REPLY to Reviewer #2

General comments. This is an interesting review on the history of rapamycin and its potential use as a de-repressor of gamma globin/fetal hemoglobin.

Answer. We thank the reviewer for her(his) positive comments

Specific comments.

Point 1. There are number of minor proofreading/grammar/usage issues, several of which are pointed out below.

Line 97: Should say “were transferred”

Line 105: Should say “rapamycin-mediated”

Line 108: Should say “others”

Line 243: Should say “pointed” instead of “spotted”

Line 278: Should say “being analyzed” instead of “under elaboration”

Line 283: Should say “influence”

Line 321: Should say “proposing”

Answer. All these suggestions have been followed. We found other minor English errors that have been fixed (red-marked changes).

Point 2. Many authors prefer to use "mechanistic Target Of Rapamycin" as the elaboration of "mTOR" rather than "mammalian", because it is similarly applicable to non-mammalian or mammalian cell studies.

Answer. We thank the reviewer for raising this point. We agree with the suggestion and we have used "mechanistic Target Of Rapamycin" as the elaboration of "mTOR" (page 3, line 121).

Point 3. On line 53, the clinical syndrome to which the authors refer is more correctly designated "hereditary persistence of fetal hemoglobin", which more accurately correspondence with the acronym HPFH.

Answer. The suggestion has been followed (page 2, line 53).

We thank the reviewer for the general positive comments and for the suggestions.
